# Cancer care disruption and reorganisation during the COVID-19 pandemic in Australia: A patient, carer and healthcare worker perspective

Rhiannon Edge[1‡], Josh Meyers[2‡*], Gabriella Tiernan[1], Zhicheng Li[1], Alexandra Schiavuzzi[1], Priscilla Chan[1], Amy Vassallo[3], April Morrow[1], Carolyn Mazariego[1,4], Claire E. Wakefield[5,6], Karen Canfell[1,4], Natalie Taylor[1,4]

1 Daffodil Centre, The University of Sydney, a joint venture with Cancer Council New South Wales, Sydney, NSW, Australia, 2 Cancer Council NSW, Sydney, NSW, Australia, 3 The George Institute for Global Health, UNSW, Sydney, NSW, Australia, 4 The University of Sydney School of Public Health, Faculty of Medicine and Health, University of Sydney, Sydney, NSW, Australia, 5 Behavioural Sciences Unit, Kids Cancer Centre, Sydney Children's Hospital, Sydney, NSW, Australia, 6 School of Women's and Children's Health, UNSW Sydney, Sydney, Australia

‡ RE and JM are co-first authors.
* Joshua.meyers@nswcc.org.au

**Data Availability Statement:** Data cannot be shared publicly because of ethics approval limitation. Data are available from the Cancer

## Abstract

The COVID-19 pandemic has dramatically impacted cancer care worldwide. Disruptions have been seen across all facets of care. While the long-term impact of COVID-19 remains unclear, the immediate impacts on patients, their carers and the healthcare workforce are increasingly evident. This study describes disruptions and reorganisation of cancer services in Australia since the onset of COVID-19, from the perspectives of people affected by cancer and healthcare workers. Two separate online cross-sectional surveys were completed by: a) cancer patients, survivors, carers, family members or friends (n = 852) and b) healthcare workers (n = 150). Descriptive analyses of quantitative survey data were conducted, followed by inductive thematic content analyses of qualitative survey responses relating to cancer care disruption and perceptions of telehealth. Overall, 42% of cancer patients and survivors reported experiencing some level of care disruption. A further 43% of healthcare workers reported atypical delays in delivering cancer care, and 50% agreed that patient access to research and clinical trials had been reduced. Almost three quarters (73%) of patients and carers reported using telehealth following the onset of COVID-19, with high overall satisfaction. However, gaps were identified in provision of psychological support and 20% of participants reported that they were unlikely to use telehealth again. The reorganisation of cancer care increased the psychological and practical burden on carers, with hospital visitation restrictions and appointment changes reducing their ability to provide essential support. COVID-19 has exacerbated a stressful and uncertain time for people affected by cancer and healthcare workers. Service reconfiguration and the adoption of telehealth have been essential adaptations for the pandemic response, offering long-term value. However, our findings highlight the need to better integrate psychosocial support and the important

Council NSW Institutional Data Access / Ethics
Committee (contact via ethics@nswcc.org.au) for
researchers who meet the criteria for access to
confidential data.

**Funding:** The author(s) received no specific
funding for this work.

**Competing interests:** The authors have declared
that no competing interests exist.

role of carers into evolving pandemic response measures. Learnings from this study could inform service improvements that would benefit patients and carers longer-term.

## Introduction

The COVID-19 pandemic has dramatically and precipitously altered the cancer care land-scape. Disruptions have been seen across all facets of cancer care, from delaying diagnoses and treatment to halting clinical trials, and diminishing access to psychosocial support [1–6]. The resilience of healthcare systems to withstand significant operational pressures whilst maintaining high quality cancer care continues to be tested.

Australia's pandemic response has largely been effective by international comparisons [7]. With a population of around 25 million, Australia has experienced lower infection and death rates than many comparable countries, with 28,978 confirmed cases and 909 deaths as of March 1st, 2021 [8]. While the Australian health system has not been adversely affected to the same extent as many other countries in Europe and North America, the pandemic necessitated extensive preparations and widespread changes to routine healthcare.

People with cancer may have increased vulnerability to COVID-19 due to their need for regular access to–oftentimes intensive and immune-suppressing–treatment and care [9]. In response, cancer treatments were modified according to rapidly published guidelines, and hospital visitation restrictions were applied [1, 3, 5, 10]. Although cancer-related services were classified as vital and remained available (albeit with modifications), utilisation declined [11, 12], with cancer screening programs temporarily paused by Australian health authorities (e.g., BreastScreen) or experiencing disruption (e.g., National Cervical Screening Program) [1]. The potential effects of reduced access to and under-utilisation of cancer-related health services are substantial. Prolonged delays in diagnoses and treatment can lead to a more advanced stage of cancer at diagnosis, poorer health outcomes, and subsequent downstream health system effects (such as greater costs to health systems and surges in diagnostic and treatment demand) [13, 14].

The pandemic and associated public health measures have also had profound impacts on the lives of people diagnosed with cancer and their support networks [6]. People with cancer and their families already experience great uncertainty about their future, which may be further exacerbated by concerns about contracting the virus, disruptions to their care, and the effects of social isolation [6]. While the long-term effects of pandemic response measures and subsequent care disruptions on the prognosis of people with cancer is not yet known, the psychosocial impact on patients, families, and carers is increasingly evident [2, 15].

Further, through a confluence of factors including physical distancing measures, patients' reluctance to visit healthcare centres, and efforts to reduce demand on acute care service, new models of healthcare delivery in the form of telehealth and video consults have emerged [5, 12]. Healthcare providers have been encouraged to offer virtual care throughout the pandemic, with the Australian Government introducing a series of new Medicare Benefits Schedule (MBS) item numbers–a list of Government subsidised services–for telehealth consults throughout 2020 [11]. Whilst telehealth had previously been adopted in some settings, including rural and remote communities [16] and specialist care [17], COVID-19 prompted a rapid scale-up (of predominantly telephone-based telehealth) in less established settings, including general practice, allied health and hospital outpatient clinics [11]. Such rapid acceleration of this digital transition is unprecedented, with limited time for preparations or extensive piloting with healthcare professionals, patients and their carers.

The aims of this study are to describe the ways in which cancer care in Australia has been disrupted and reorganised during the COVID-19 pandemic, and to understand the impact this has had from the perspectives of both people affected by cancer and healthcare workers.

## Methods

### Participants and design

Two concurrent cross-sectional surveys were conducted online to assess the impact of COVID-19 on cancer care delivery in Australia with: a) people affected by cancer, including cancer patients, cancer survivors, carers, family members or friends and b) healthcare workers. People affected by cancer were eligible to participate if they resided in Australia, were aged 18 or above, and had a current or previous diagnosis of cancer, or were a caregiver, family member or friend of someone diagnosed with cancer. Respondents were asked tailored survey questions depending on their relationship with the person affected by cancer (e.g., caregiving frequency, attendance at appointments). Healthcare workers were eligible to participate if they were employed or involved in the delivery of cancer services in Australia.

The surveys were advertised on the Cancer Council NSW and Cancer Council Australia websites and were promoted via social media (Facebook and Twitter). Invitations were also distributed through existing cancer community and healthcare professional networks across Australia (e.g., Cancer Voices and Clinical Oncology Society of Australia). All survey responses were collected between 22 June and 30 September 2020. During this time, Australia was under nation-wide COVID-19 restrictions and both international and inter-state travel bans were in place. The study period also corresponded with the COVID-19 cluster which emerged in Melbourne, Victoria in late June 2020, resulting in more than 110 days of lockdown restrictions. Under these restrictions, Melbournians and other Victorian residents were ordered to stay at home unless they needed to leave for medical reasons, essential shopping, essential work, or outdoor exercise (within geographical limits). Upon closure of the survey on 30 September 2020, Australia had recorded 27,000 COVID-19 infections and 886 deaths [18].

Ethical approval was granted for this research by Cancer Council NSW's Human Research and Ethics Committee (HREC) (Ref: #322). For both the surveys, participants were directed to an online version of the Participant Information Statement and specifically asked for their consent before beginning the survey.

### Measures

Demographic data and other descriptive information were collected for all participants. For those affected by cancer, data were collected on year of diagnosis, cancer stage, cancer type and current treatment. For healthcare workers, data were collected on the type of profession, length in role, primary work setting, and whether their role involved direct clinical interactions with patients.

Both surveys comprised of sections regarding changes to cancer care delivery and experiences of telehealth. The measures used in the surveys were based on available pre-existing evidence around the impact of COVID-19, adapted versions of relevant validated tools, and stakeholder (i.e., clinician, consumer) consultation. For the measures used in the survey for those affected by cancer, questions were informed by an in-depth qualitative analysis of a National Cancer Information and Support Line and Online Cancer Community Forum in Australia (https://www.cancer.org.au/online-resources/cancer-council-online-community) [19]. Additionally, some measures regarding cancer care quality and telehealth described in this paper were adapted from The Cancer Care Coordination Questionnaire for Patients [20] and The Telehealth Satisfaction Scale [21]. The healthcare worker survey included a specific

section focused on cancer care delivery and reorganisation which is the focus of this analysis. The development of measures in this section of the survey were informed by emerging evidence regarding the impact of COVID-19, the literature on previous crisis events (e.g., the Severe Acute Respiratory Syndrome (SARS) epidemic), and healthcare performance. Four key categories focussed on exploring quality of care [22], organisational support [e.g., 23], resources [24], time and priorities [25], in consideration of the crisis. In both surveys, our definition of telehealth included both telephone and online consults.

The study surveys were tested by our consumer representative and by reviewers experienced in cancer care to assure quality while piloting was not done due to the time critical nature of the survey. Data quality was checked at regular intervals post survey release.

## Data analysis

Demographic data and quantitative survey responses were examined using frequency and percentages for categorical variables, and mean and standard deviation for continuous variables. Qualitative survey responses were imported into NVivo 12 Plus software (QSR International Pty Ltd 2018) to separately undergo an inductive thematic qualitative content analysis. Using this approach, a preliminary review of the content was performed by GT to develop the initial coding structure which was then shared with the research team (RE, JM and NT) for secondary review and comment. This informed the refinement of the coding framework which was then applied to the complete dataset. Similar codes were grouped together into higher order categories and subthemes, identified by salience and frequency [26]. In the results, alignments between quantitative survey findings and the emergent themes from the qualitative analysis were identified and reported.

## Results

A total of 852 people affected by cancer (683 cancer patients and survivors; and 169 carers, family members and/or friends) and 150 healthcare workers participated in the two surveys. Participants' demographics are shown in **Table 1.** In both surveys, more than 50% of the sample provided at least one response to an open-ended question relating to cancer care disruption and telehealth.

Results focus on four common areas surveyed with healthcare workers and those affected by cancer to understand the impact of cancer service disruption and reorganisation from different perspectives: 1) experiences of cancer service and care disruption, 2) the perceived quality of clinical care during the pandemic, 3) experiences of telehealth, and 4) the impacts of COVID-19 on the supporting role of carers and family members in cancer care. Descriptive quantitative data from the survey is accompanied by qualitative findings and representative quotes to illustrate key themes.

### Experiences of cancer service and care disruptions

During the pandemic, a range of disruptions to routine cancer care services were reported by both those affected by cancer and healthcare workers. Overall, 42% of cancer patients and survivors reported experiencing some level of disruption to their cancer care or treatment. Of this cohort, 44% (n = 118) were receiving active treatment (including chemotherapy, radiation therapy, immunotherapy or surgery). 49% of carers suggested there had been care disruptions for the person whom they cared for. More specifically, almost a third of cancer patients and survivors (28%) indicated that their medical appointments had needed to be rescheduled, 10% reported that their cancer treatment had been postponed, 7% reported elective surgeries had

**Table 1. Participants' characteristics.**

| Demographic characteristics | People affected by cancer | | Healthcare workers |
| --- | --- | --- | --- |
| | Cancer patients and survivors (n = 683) n (%) | Carers, family members or friends (n = 169) n (%) | (n = 150) n (%) |
| **Age** | | | |
| 18–24 years | 1 (0.1) | 0 (0.0) | 4 (2.7) |
| 25–34 years | 13 (1.9) | 9 (5.3) | 33 (22.4) |
| 35–44 years | 38 (5.6) | 27 (16.0) | 33 (22.4) |
| 45–54 years | 109 (16.0) | 36 (21.3) | 30 (20.4) |
| 55–64 years | 206 (30.2) | 46 (27.2) | 37 (25.1) |
| 65–74 years | 230 (33.7) | 32 (18.9) | 8 (5.4) |
| 75+ years | 86 (12.6) | 19 (11.2) | 2 (1.3) |
| **Gender** | | | |
| Woman | 477 (69.8) | 136 (80.5) | 121 (81.2) |
| Man | 204 (29.9) | 33 (19.5) | 27 (18.1) |
| Prefer not to say | 2 (0.3) | 0 (0.0) | 1 (0.7) |
| **Region** | | | |
| Metropolitan | 429 (62.8) | 98 (58.0) | 129 (86.6) |
| Rural or remote | 254 (37.2) | 71 (42.0) | 20 (13.4) |
| **State** | | | |
| New South Wales | 529 (77.5) | 134 (79.3) | 94 (63.1) |
| Victoria | 59 (8.6) | 13 (7.7) | 33 (22.1) |
| Northern Territory | 1 (0.1) | 0 (0.0) | 1 (0.7) |
| Queensland | 31 (4.5) | 11 (6.5) | 5 (3.4) |
| South Australia | 15 (2.2) | 1 (0.6) | 4 (2.7) |
| Western Australia | 15 (2.2) | 2 (1.2) | 7 (4.7) |
| Australian Capital Territory | 12 (3.1) | 4 (2.4) | 4 (2.7) |
| Tasmania | 21 (1.8) | 4 (2.4) | 1 (0.7) |
| **Time since the most recent cancer diagnosis (in years)** | | | |
| Mean ± Standard  Deviation (SD) | 5.1 ± 6.9 | 2.9 ± 4.4* | _ |
| **Cancer stage** | | | |
| Recovery or remission | 233 (34.1) | 15 (12.0)* | _ |
| Early | 144 (21.1) | 18 (14.4)* | _ |
| Localised | 100 (14.6) | 13 (10.4)* | _ |
| Regional spread | 53 (7.8) | 13 (10.4)* | _ |
| Distant spread or metastatic | 95 (13.9) | 51 (40.8)* | _ |
| Unsure | 58 (8.5) | 15 (12.0)* | _ |
| **Cancer type**** | | | |
| Breast | 259 (38.1) | 19 (14.7)* | _ |
| Skin Cancer, inc. melanoma | 88 (13.0) | 11 (8.5)* | _ |
| Prostate | 79 (11.6) | 14 (10.9)* | _ |
| Lymphoma | 56 (8.2) | 12 (9.3)* | _ |
| Colorectal | 53 (7.8) | 10 (7.8)* | _ |
| **Cancer treatment** | | | |
| Chemotherapy | 158 (23.3) | 54 (32.0)* | _ |
| Radiotherapy | 87 (12.9) | 28 (16.6)* | _ |
| Immunotherapy | 67 (9.9) | 13 (7.7)* | _ |
| Surgery | 106 (15.7) | 18 (10.7)* | _ |
| Anti-hormone | 121 (17.9) | 7 (4.1)* | _ |

(*Continued*)

**Table 1.** (Continued)

| Demographic characteristics | People affected by cancer | | Healthcare workers |
|---|---|---|---|
| | Cancer patients and survivors (n = 683) n (%) | Carers, family members or friends (n = 169) n (%) | (n = 150) n (%) |
| Active surveillance | 176 (26.0) | 17 (10.1)* | – |
| Completed treatment | 203 (30.0) | 16 (9.5)* | – |
| Other | 43 (6.4) | 25 (14.8)* | – |
| **Relationship to the person with cancer** | | | |
| Partner or spouse | – | 62 (36.7) | – |
| Child | – | 45 (26.6) | – |
| Other | – | 62 (36.7) | – |
| **Cohabit with the person with cancer** | | | |
| Yes | – | 87 (51.5) | – |
| No | – | 82 (48.5) | – |
| **Primary work setting** | | | |
| Specialist/dedicated cancer treatment centre | – | – | 27 (32.9) |
| Inpatient hospital | – | – | 51 (34.2) |
| Outpatient service | – | – | 33 (22.1) |
| Other | – | – | 38 (25.5) |
| **Healthcare role** | | | |
| Medical Doctor (non-GP) | – | – | 27 (18.0) |
| Nurse | – | – | 40 (26.7) |
| Allied Health | – | – | 37 (24.7) |
| Clinical Trial Coordinator | – | – | 21 (14.0) |
| General Practitioner | – | – | 4 (2.7) |
| Other | – | – | 21 (14.0) |
| **Direct clinical interaction with patients** | | | |
| Yes | – | – | 129 (86.0) |
| No | – | – | (14.0) |

*Caregivers have reported cancer information on behalf of the person they care for.

**Five most common cancer types only shown in Table 1.

been delayed, and 12% suggested their planned cancer screening tests had been rescheduled (**Fig 1**).

Cancer patients and survivors described feelings of stress and anxiety associated with these rapidly unfolding changes, and highlighted communication issues with their healthcare provider in some instances:

> "[My] annual check-up with the surgeon was postponed for the duration. I had my annual mammogram prior to this. . . had to assume that the results were OK as no-one contacted me." (Survivor)

> "Due to social distancing the <hospital name removed> put my treatment on hold. This has been very stressful." (Patient)

Although reported cancellations were relatively low (suggested by 10% of respondents overall), there were still instances where appointments, surgeries, clinical trials and hospital

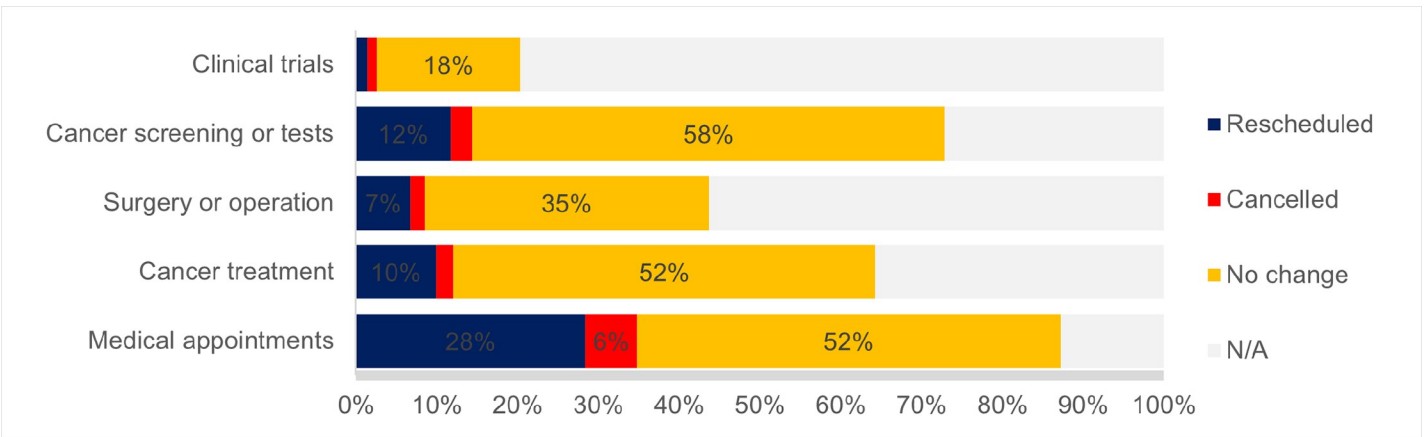

**Fig 1. Disruptions to cancer care reported by cancer patients and survivors**\*. Based on answers to the survey item: 'Have you experienced any of the following changes to your cancer care since the COVID-19 outbreak in Australia?' (n = 683). \*N/A responses were included in percentage calculations, missing data were excluded.

support groups or programs had been cancelled due to the reorganisation of cancer care in response to the pandemic.

> *"With our hospital closure, my treatment plan disappeared."* (Patient)

> *"The wellness programs were cancelled which impacted me both socially & physically."* (Patient)

> *"Cancellation of all support groups face to face. Cancellation of hospital wellness programs."* (Patient)

Patients and carers also articulated frustrations about the perceived prioritisation of COVID-19 over their cancer-related symptoms and reported that this caused distress. Some patients also described their hesitancy to attend future appointments at healthcare centres due to their fears of potential COVID-19 exposure.

> *"Having to have a COVID test because of symptoms that are caused by treatment is a complete waste of time and unnecessarily distressing."* (Carer)

> *"[I have] anxiety at medical check-ups when staff need to be very close e.g., biopsy, examinations. Less likely to see GP face to face for minor issues e.g., skin issues which cannot be dealt with via telehealth."* (Patient)

> *"[I am] fearful of attending appointments for X-rays, ultrasounds even if they are only yearly follow ups for my breast cancer."* (Survivor)

Disruptions to cancer services were also reported by healthcare workers, with 43% agreeing that there had been atypical delays in delivering cancer care, and half (50%) agreeing that patient access to research and clinical trials had been reduced since the onset of the pandemic. In associated comments, they provided details about delays to essential cancer diagnostic procedures and treatments:

> *"Non-essential surgery was cancelled/delayed. I did not directly deal with the consequences of that with most patients but nursing and psychology staff did."* (Healthcare worker)

*"Transplants were made more difficult due to trouble precuring donors and getting cells, more organisation time, more delays and sometimes no available donor."* (Healthcare worker)

Further, consistent with the hesitancy reported by some cancer patients and survivors in attending healthcare centres, 45% of healthcare workers agreed that the number of consultations cancelled by patients had increased from prior to the pandemic. They expressed concern about the potential long-term impacts of reduced appointment attendance and delays to treatment and diagnostics on long-term cancer prognosis:

*"Very concerned about the reduction in diagnosed cases and what that will do to our patients and us over the next 5 years."* (Healthcare worker)

*"[It is] concerning for staff that cancer referrals are well down on usual and so we worry about more advanced presentations and our workload over next few months."* (Healthcare worker)

## Perceived quality of clinical care during the pandemic

Despite the disruptions observed across cancer services during the study period, 61% of cancer patients, survivors and carers agreed that they felt fully informed about changes to cancer treatment plans. Further, the majority reported that they did not encounter difficulties obtaining appointments with their regular healthcare provider, nor experience longer wait times for appointments (Fig 2). This sentiment was also echoed in some of the comments they provided in the survey:

*". . .I have had very good GP and specialist support and have had no trouble accessing their services through their clinics."* (Patient)

*"I feel I have received exceptionally good care from the radiation therapists, nurses and specialists (particularly during face-to-face consults) throughout my cancer treatment. I had surgery in early April and completed 5 weeks of radiotherapy yesterday."* (Patient)

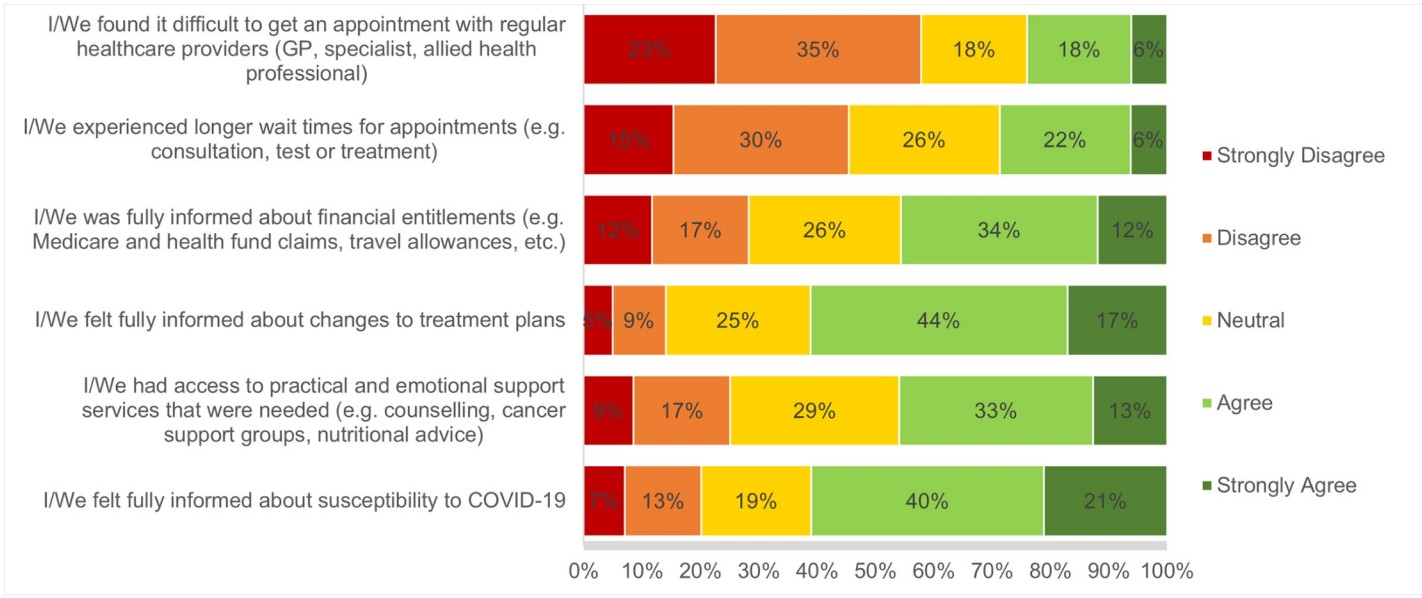

**Fig 2. Perceptions of cancer care quality during COVID-19 from cancer patients, survivors and carers**\*. Based on answers to the survey item: 'To what extent do you agree or disagree with the following statements about your/their cancer care experience during the COVID-19 outbreak?'. \*N/A responses and missing data were excluded from percentage calculations.

However, whilst the majority responded positively, there were also some concerns flagged by patients and carers about care quality. In some instances, they described a lack of information provided by the health system about service disruptions, particularly in the earlier weeks of the pandemic. Further, some explained how their cancer treatment plans had essentially "*disappeared*" and that they were experiencing challenges in accessing usual care.

> *"The hospital just carried on as if nothing had changed for the first 6 weeks of the pandemic. There was no additional information given."* (Carer)

> *"Would like some information about the resumption of non-emergency surgery."* (Patient)

For healthcare workers, similar ambiguities to patients and carers were identified in their perceptions of care quality during COVID-19. As demonstrated in **Fig 3**, 72% reported being happy with the quality of care they had been able to provide to cancer patients during this crisis, however, almost a third (31%) suggested that changes to cancer diagnostic procedures and/ or pathways had made them suboptimal. Healthcare workers highlighted the additional challenges of maintaining quality of care and meeting patients' needs during the pandemic, with 73% agreeing that the complexity of care had increased and 85% agreeing (50% strongly agreeing) that COVID-19 had increased pressure on their patients' mental health and wellbeing.

> *"Definitely noticed patients undergoing cancer treatment feel vulnerable to severe COVID infection and are more likely to go to extreme measures to isolate with subsequent increase in anxiety/depression/loneliness/job loss/financial and relationship strain."* (Healthcare worker)

> *"COVID-19, and the associated impact of this has been an overlay that has affected every aspect of work—more complex patient interactions, more complex workplace interactions (and my own personal capacity to deal with challenges has been reduced)."* (Healthcare worker)

## Experiences of telehealth during the pandemic

Prior to the pandemic, only 17% of cancer patients, survivors and carers reported that they had used telehealth services for their cancer care, compared to nearly three quarters (73%) after the onset of COVID-19.

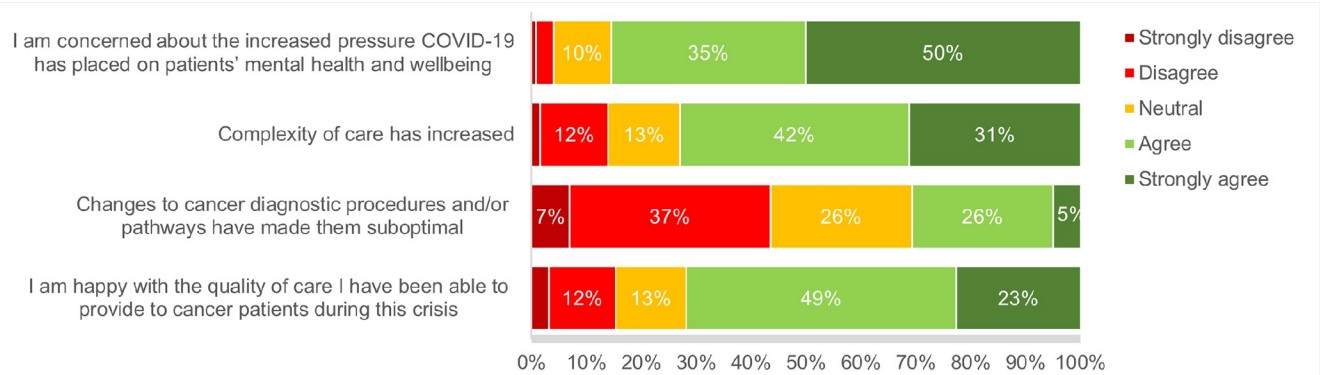

**Fig 3. Health worker perceptions of cancer care quality during COVID-19**\*. Based on answers to the survey item: 'To what extent do you agree or disagree with the following statements about cancer care during the COVID-19 outbreak?'. \*N/A responses and missing data were excluded from percentage calculations.

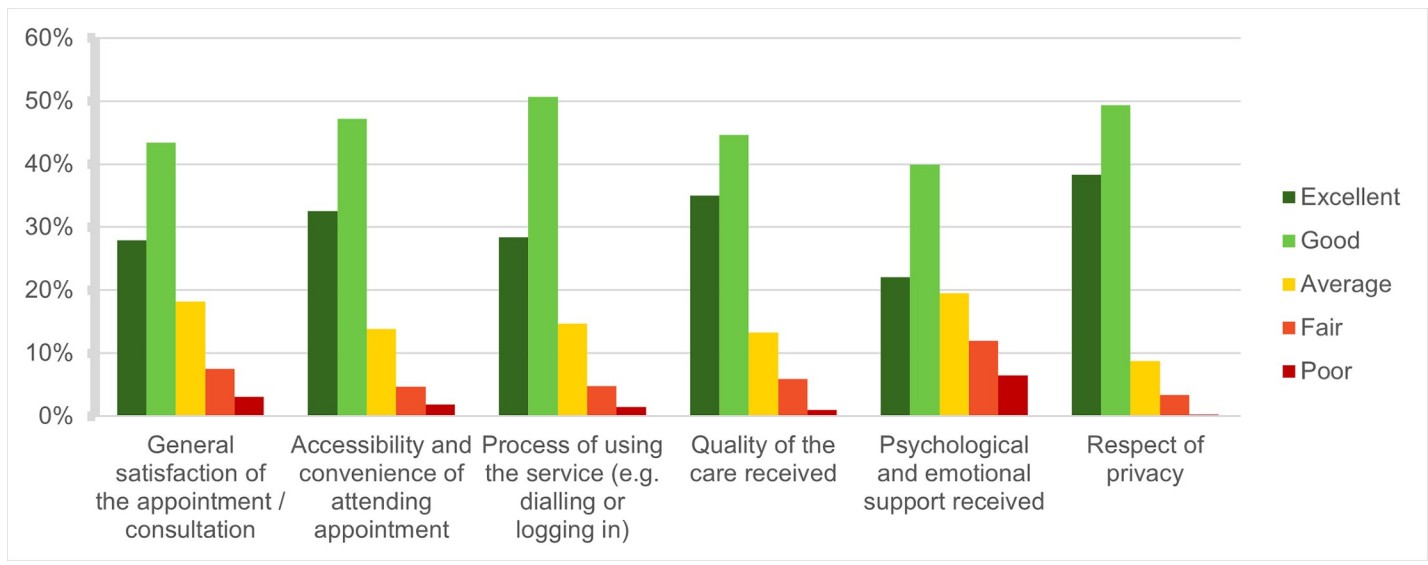

**Fig 4. Experiences of telehealth for cancer patients and survivors during COVID-19**[*]. Based on answers to the survey item: 'Please rate your overall experience of the telehealth services you received'. [*]N/A responses and missing data were excluded from percentage calculations.

Cancer patients and survivors indicated that most of these telehealth consults during the pandemic had been with their GP (65%) or a specialist consultant (64%) (e.g., medical oncologist). A smaller proportion had received psychological support or had an appointment with an allied health professional over telehealth. For healthcare workers, 60% reported that more than half of their regular appointments had been moved to telehealth consultations since the onset of the pandemic, with the majority (68%) agreeing they had received adequate support to adopt these alternative modes of service delivery.

Overall, general patient satisfaction with telehealth consults for their cancer care was high, with 80% rating the accessibility and convenience of the appointment and quality of care they received as either 'good' or 'excellent'. Respect of privacy and the process of logging/dialling into the appointment were also highly rated. However, when asked about the psychological and emotional support they had received over telehealth, 18% reported their experience had been 'poor' or 'fair' (**Fig 4**).

The main perceived benefits of telehealth for cancer patients and survivors included the reduced exposure risk to COVID-19, reduced costs of attending treatment (i.e., travel, parking, childcare) and more timely access to services. In contrast, the most frequently cited difficulties were concerns about whether telehealth would deliver the same quality of care as a physical examination, with 18% of respondents suggesting this was of concern for them. Cancer patients, survivors and carers described a lack of personalised care (e.g., reduced rapport and interpersonal connection) and difficulties interpreting information they had been provided through online or telephone consults. They also highlighted how telehealth was not always optimal for certain types of appointments, particularly when physical examinations were required, or psychological support was being delivered. When cancer patients and survivors were asked about their likelihood of using telehealth services in future, more than half (55%) indicated that they would use this again, however, 1 in 5 (20%) answered that they were unlikely to.

Healthcare workers also articulated benefits of telehealth such as decreased wait times, streamlined processes and increased privacy. However, they too raised concerns about the appropriateness of telehealth consultations where physical assessments were required, or

interpersonal connection was important for the delivery of optimal care. Healthcare workers also described the challenges they had experienced with making accurate diagnoses over telehealth and highlighted how non-verbal communication and body language cues that may signify psychological distress were easier to miss. The positive and negative perceptions of both healthcare workers and patients or carers in relation to telehealth are shown in **Table 2**.

## Impacts of the pandemic on the supporting role of carers and family members in cancer care

Over three quarters (79%) of healthcare workers reported that involving carers in consults had become more challenging since the onset of the COVID-19 pandemic. They explained how the restrictions on hospital visitation had been difficult to communicate to patients and their families, and that the reduction in essential support was likely having a negative impact on their mental health and wellbeing.

> *"Sad for patients that all visitors have been banned from the wards—patients missing out on so much support from family and friends."* (Healthcare worker)

> *"Families do find it challenging, understandably, due to restrictions on how many people can visit. . ."* (Healthcare worker)

Patients and their carers similarly described the challenges they faced attending in-person consults together as a result of restricted hospital visitation and social distancing measures. Further, almost half (48%) of carers reported that they had been unable to attend telehealth consults with the person diagnosed by cancer. They described how not being able to attend appointments–whether in-person or via telehealth–had placed an additional burden on them to provide the same levels of practical and emotional support as before the pandemic. Some carers suggested that they were no longer able to listen to and clarify important information for their loved ones, nor understand how they were best able to provide support during intensive treatments. Patients also highlighted the lonely experience of attending appointments with a reduced social support network. They felt overwhelmed attending cancer treatments alone, and described challenges with digesting and interpreting health information from clinicians without this additional support.

> *"Because I haven't been able to attend appointments (which I would have previously) I haven't been able to provide mum and dad with support and clarification. This has placed an additional burden on dad as he has had to make sense of the information and then communicate it to me."* (Carer)

> *"It was also lonely attending appointments alone and not being able to take a support person to radiotherapy on occasions."* (Patient)

> *". . .while I'm always afraid, I am more afraid as to the treatment as I think my husband won't be allowed [to attend the appointment]."* (Patient)

> *"Made my ability to have the same sort of support from family friends very difficult during hospital stays and at home support post surgery."* (Patient)

## Discussion

Although Australia's response to the pandemic has largely been effective in terms of limiting case numbers and fatality rate [7], disruptions to cancer services, as outlined in this research

**Table 2. Positive and negative perceptions of telehealth reported by those affected by cancer and healthcare workers.**

| Positive perceptions | | Negative Perceptions | |
|---|---|---|---|
| **Patients, Survivors, Carers or Family Members** | | | |
| *Improved accessibility and convenience* | "[Telehealth] saves heaps of time." (Patient) | *Appropriateness of telehealth for certain types of consults* | "Medical follow up appointment changed to teleconference, which was not really appropriate for the type of appointment (lymphoedema review)." (Carer) |
| | "[Telehealth] appointment was on time versus 4-hour wait when last at the same public hospital when face-to-face appointment." (Patient) | | "The end of life care in <facility name removed> transitional care program was not good due to the cancer/palliative care service working 'remotely' I.e. talking to her on the phone, not going to visit her—completely unacceptable." (Carer) |
| | "We also use phone consultations with medical staff which are great, much better than travelling to appointments." (Patient) | | "Switch to phone consults with specialist and no recording of vitals such as weight because no face to face with clinic nurses. [This] led to delay in identifying food intake problem, and backlog of endoscopy procedures led to dangerous delay in treating problem." (Carer) |
| | "[patient's] increased lack of mobility over the last couple of weeks means that telehealth is his only option." (Carer) | *Difficulties interpreting health information* | "I didn't fully understand all side effects of suggested medicine over the phone." (Patient) |
| *Respect of privacy* | "...more privacy when taking calls for telehealth." (Carer) | *Concerns about care quality vs face to face consults* | "I prefer speaking face-to-face for first appointments... then I feel the doctor or support person has an understanding of my personality type." (Patient) |
| *Reduced COVID-19 exposure risk* | "I had a teleconference call with my cancer doctor, so I wasn't put at any risk and can email her at any time." (Patient) | | "Mum probably could have benefited from a social worker or someone similar who could have 'seen' she is not doing okay. This is hard to pick up over the phone." (Carer) |
| **Healthcare Workers** | | | |
| *Improved quality of care* | "I hope that telehealth will be an ongoing form of healthcare as it is great for some patients e.g. for follow ups, sorting out issues without frail, elderly people needing to come into hospital multiple times." | *Concerns about care quality vs face to face consults* | "Since I am not seeing the Day Oncology patients face-to-face anymore my ability to accurately assess them has decreased. Patients appear more positive over the phone and often the accurate picture of how bad they are doing is not identified." |
| *Respect of Privacy* | "I am working from home & finding phone consults more effective than consults in busy oncology settings with no privacy & especially now with the need for PPE & social distancing." | | "Remote consultations (video or phone) make it difficult to accurately assess signs and symptoms, as well as making it difficult to detect cues for distress. Relying on patient self-report may be sub-optimal." |
| *Improved accessibility and convenience* | "It has been fantastic to have telehealth in the crisis as it has meant that over resourced, stupid and clunky practices have had to be removed-hopefully never to return. Patients, in the main, have been wonderfully adaptable and accepting of the situation. No-one wants to sit around for up to 3 hours in an outpatient department waiting and waiting and waiting." | | "Whilst telehealth has been a good option rather than nothing, nothing beats a face-to-face appointment where you can observe the unsaid and follow up on these cues for better holistic cancer care" |
| *General satisfaction* | "Telehealth may be useful tool to continue." | | "More difficult to build a rapport & have the human connection as doing 95% of consults via phone." |
| | | *Issues with technology* | "The majority 80% plus of my patients in the community have no idea how to use technology to maximise benefit to themselves. Even younger people that you may think could use technology are struggling." |
| | | *Issues with technology/ Concerns about care quality vs face to face consults* | "...extended telehealth delivery of services has been challenging in Psychology. Working with older adults has been challenging as many are unable to operate video conferencing and completing assessments via phone means that non-verbal information is missing. There are a number of patients that I have now never met face to face." |

and elsewhere [2, 6, 11, 12], demonstrate an immediate psychosocial impact on people affected by cancer and may have longer term implications.

Our findings highlight that patients, survivors, carers, and healthcare workers have adapted relatively well to unprecedented overhauls in standard protocols and offer important insights for future policy and practice in relation to crisis preparedness, as well as ongoing cancer care. During our study period, disruptions were observed across a wide range of cancer services, including screening, diagnostics and treatments, and consultation formats, aligning with other recent research [11, 12]. Whilst the true cost of these disruptions will take time to understand, there are concerns that such rapid health service reconfiguration will result in delayed diagnosis and poorer prognosis [10, 27]. Using breast cancer as an example, modelling commissioned by the Australian Government suggests temporary pauses to BreastScreen services could adversely impact clinical outcomes including time to diagnosis and survival (a potential reduction in population-level survival rates of up to 1.9% by end 2023) [28]. Additionally, modelling of the 6.5-month period of COVID-19-related restrictions in Victoria–Australia's most affected state–found a 10% reduction in cancer pathology notifications and an estimated 2,530 missed or delayed cancer diagnoses [27]. Furthermore, international evidence suggests that for many cancers, even modest delays to treatment can result in a substantial proportion of patients with early-stage cancers progressing from having curable to incurable disease [29]. Ongoing work by the COVID-19 and Cancer Global Modelling Consortium (CCGMC) is quantifying the impact of delays on outcomes, in addition to disruptions to treatment of cancer [30]. Post-pandemic service planning must consider these concerns to minimise the unintended consequences of the COVID-19 response. Given the key role of primary care in cancer screening and diagnosis, strategies to address potential diagnostic and treatment queues will require collaboration across all sectors of the health system.

Considering the immediate and widespread disruptions experienced by people with cancer, it is a positive reflection on the Australian pandemic response effort that patients and carers perceived changes to their cancer treatment plans were communicated effectively and wait times were not adversely affected for the majority. Indeed, only a small proportion of healthcare workers perceived changes to cancer care to be sub-optimal. However, the psychological impact of COVID-19 should not be underestimated. Given the high levels of uncertainty about future restrictions, the increasing clinical complexity of caring for people affected by cancer, and reduced social support networks, it is not surprising that healthcare workers in this study were concerned about the mental health burden of COVID-19. Recently developed guidelines highlight practical strategies for clinicians to employ with patients, caregivers, and family to address the uncertainty associated with their care and the adoption of these should be strongly encouraged [5, 10, 31]. Moreover, at a time when demand for supportive care services has surged, many community support organisations that play an important role in addressing the unmet supportive care needs of people affected by cancer have been forced to reduce their program and service offerings due to funding deficits [32]. This "perfect storm" may compound the adverse psychological outcomes of the pandemic for this vulnerable group and warrants further attention.

Not all indirect consequences of the pandemic have been undesirable. The rapid implementation of telehealth within cancer services–the adoption of which has previously been slow and fragmented across jurisdictions–has largely been successful and may offer permanent value in enhancing cancer care quality and access, as well as provide innovative and highly acceptable solutions to crisis preparedness and response strategies [33]. Telehealth has enabled healthcare workers to work remotely while patients and carers can remain at home, thereby minimising risk of virus transmission. It is important to note that despite the shift to telehealth being met with high overall levels of satisfaction among these participants, important gaps remain,

particularly in relation to perceptions of care quality, personalised care, the role of carers and the delivery of psychosocial support–echoing themes previously identified in existing tele-health literature [33, 34]. Furthermore, MBS analysis reveals that most consultations are being done by telephone rather than video telehealth [11]–despite Department of Health recommendations that videoconference services are the preferred substitute for face-to-face consultations [35]. Encouragingly, steps have already been taken to prompt proactive discussion and action on issues of trust, isolation and disconnectedness between clinicians and consumers using tele-health [5, 10]. Our findings also highlight that greater attempts to consider carers in virtual models of care should be made. Clear governance, policies and procedures to guide safety and quality in cancer telehealth consultations are clearly needed. Considering the potential value of telehealth, further research and investment from State and Federal governments is needed to address the concerns of cancer patients and carers–up to 20% of whom in our study report they are unlikely to use again–to ensure the continued viability of telehealth to augment face-to-face care.

Finally, whilst our study highlights the rapid reconfiguration of cancer care in Australia during COVID-19, further research is required to understand the extent to which such changes are clinically appropriate, offer high-value care for patients, survivors and their carers, and impact cancer outcomes. Consideration of shared decision making and patient-centred care, and evaluation of the risks and benefits of clinical decision-making–particularly as COVID-19 vaccines are rolled out for cancer patients [36]–should underpin cancer care policy and practice [5, 10, 31], both during the pandemic and beyond.

## Limitations

We acknowledge that our study has limitations. Our sample of participants recruited using national websites, existing cancer community networks and social media channels may have resulted in a convenience sample that was not representative of the wider population of those affected by cancer. We also acknowledge that certain demographic groups were overrepresented in our sample (e.g., NSW respondents, women, and people with early stage and localized cancers, and in recovery and remission). Additionally, given that only a small proportion of respondents affected by cancer were from Victoria (<10%), the state most affected by COVID-19 restrictions during the study period, our findings may underrepresent the true impacts of the pandemic on cancer care nationally.

Our study found reports of notable disruptions to cancer care, yet we could not determine the clinical appropriateness of such changes, nor the extent to which these reports corroborate objectively measured service disruption, or whether these were likely to affect morbidity and mortality. Despite these limitations, we believe that this research provides a timely and valu-able contribution to the emerging evidence of the impact of COVID-19 pandemic on cancer care, with a unique approach of exploring the experiences and perceptions of both the care recipients and care providers concurrently. The relevant timing of our two surveys and the large sample size of people affected by cancer also increases the relevance and applicability of our study. As the pandemic evolves over time, further research is required to understand the changing experiences of people affected by cancer and healthcare workers in response to the pandemic-related policy and practice changes and as COVID-19 vaccine uptake increases.

## Conclusion

The exceptional circumstances of the COVID-19 pandemic have altered cancer care. Although Australia has been largely successful in curbing the spread of the virus to date, the pandemic has exacerbated what is already an immensely stressful and uncertain time for people affected

by cancer. The long-term effects remain to be seen [6]. The reorganisation of cancer care and the adoption of telehealth has been essential, and the value of these adaptations for the pandemic response, preparedness, and beyond, is clear. However, our findings highlight the need for efforts to better integrate psychosocial support and the important role of carers into evolving pandemic response measures to guide health systems towards an equitable and person-centred digital future. Although the COVID-19 response is ongoing and contexts are constantly evolving, how we respond is ultimately dependent on how well we efficiently translate lessons learned into effective policy and practice.

## Supporting information

**S1 Survey.**
(PDF)

**S2 Survey.**
(PDF)

**S3 Survey.**
(DOCX)

**S4 Survey.**
(DOCX)

## Acknowledgments

The research team would like to thank our research consumer, Kathryn Leaney, and staff at Cancer Council NSW for their contributions to the survey question development, support with participant recruitment, and for the in-kind dedication of their time and resource to assist the research team with this project. They would also like to thank the many people affected by cancer and healthcare workers who participated in this study to provide timely insights about cancer care during the COVID-19 pandemic.

## Author Contributions

**Conceptualization:** Natalie Taylor.

**Data curation:** Gabriella Tiernan, Zhicheng Li.

**Formal analysis:** Rhiannon Edge, Gabriella Tiernan, Zhicheng Li.

**Methodology:** Rhiannon Edge, Alexandra Schiavuzzi.

**Project administration:** Priscilla Chan, April Morrow.

**Writing – original draft:** Rhiannon Edge, Josh Meyers.

**Writing – review & editing:** Rhiannon Edge, Josh Meyers, Gabriella Tiernan, Zhicheng Li, Priscilla Chan, Amy Vassallo, April Morrow, Carolyn Mazariego, Claire E. Wakefield, Karen Canfell, Natalie Taylor.

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
