## [Decision Letter · Decision Letter 0]

1 Sep 2021

Cancer care disruption and reorganisation during the COVID-19 pandemic in Australia: a patient, carer and healthcare worker perspective

PONE-D-21-09913

Dear Dr. Meyers,

We’re pleased to inform you that your manuscript has been judged scientifically suitable for publication and will be formally accepted for publication once it meets all outstanding technical requirements.

Kind regards,

Mark Wing Loong Cheong, PhD

Academic Editor

PLOS ONE

Journal Requirements:

1. Please provide additional details regarding participant consent. In the ethics statement in the Methods and online submission information, please ensure that you have specified what type you obtained (for instance, written or verbal, and if verbal, how it was documented and witnessed). If your study included minors, state whether you obtained consent from parents or guardians. If the need for consent was waived by the ethics committee, please include this information.

For additional information about PLOS ONE ethical requirements for human subjects research, please refer to " ext-link-type="uri" xlink:type="simple">http://journals.plos.org/plosone/s/submission-guidelines#loc-human-subjects-research."

2 .Please include additional information regarding the survey or questionnaire used in the study and ensure that you have provided sufficient details that others could replicate the analyses. For instance, if you developed a questionnaire as part of this study and it is not under a copyright more restrictive than CC-BY, please include a copy, in both the original language and English, as Supporting Information. Moreover, please include more details on how the questionnaire was pre-tested, and whether it was validated. 

Reviewers' comments:

Reviewer's Responses to Questions

**Comments to the Author**

1. Is the manuscript technically sound, and do the data support the conclusions?

Reviewer #1: Yes

Reviewer #2: Yes

2. Has the statistical analysis been performed appropriately and rigorously? 

Reviewer #1: Yes

Reviewer #2: Yes

3. Have the authors made all data underlying the findings in their manuscript fully available?

Reviewer #1: No

Reviewer #2: Yes

4. Is the manuscript presented in an intelligible fashion and written in standard English?

Reviewer #1: Yes

Reviewer #2: Yes

5. Review Comments to the Author

Reviewer #1: Cancer care disruption during the COVID-19 pandemic is very important issue that the authors highlighted well.

It is such a shame that Melbournians/Victorians are underrepresented in this study. It would have been very useful if we could have seen those results by State as the impact of COVID 19 was different across Australia and Victoria was the only state that was experienced stage 4 harsh lock down for 110 days. We also know that men have lower survival from most cancer types than women. So, the effect of COVID 19 might be more for male cancer patients.

I think the authors addressed the research question well considering the limitations of the data. Hopefully, future studies could shed more light on this important issue. I think these results are tip of an iceberg, definitely underestimated the real impact of the pandemic on cancer diagnosis, treatment and palliative care.

Reviewer #2: Dear Editor,

The COVID-19 pandemic has dramatically impacted cancer care worldwide.

Disruptions have been seen across all facets of care. While the long-term impact of

COVID-19 remains unclear, the immediate impacts on patients, their carers and the

healthcare workforce are increasingly evident. This study describes disruptions and

reorganisation of cancer services in Australia since the onset of COVID-19, from the

perspectives of people affected by cancer and healthcare workers.

This paper well describe actions taken in place to fight service disruption due to covid pandemic.

. ata from the paper can aid to build national programs to avoid service disruption during pandemic. It is well conceived and written and we recommend publication. No revision should be provided by authors.

6. PLOS authors have the option to publish the peer review history of their article (what does this mean?). If published, this will include your full peer review and any attached files.

Reviewer #1: No

Reviewer #2: No

---

## [Editor Report · Acceptance letter]

9 Sep 2021

PONE-D-21-09913 

Cancer care disruption and reorganisation during the COVID-19 pandemic in Australia: a patient, carer and healthcare worker perspective 

Dear Dr. Meyers:

I'm pleased to inform you that your manuscript has been deemed suitable for publication in PLOS ONE. Congratulations! Your manuscript is now with our production department. 

Kind regards, 

on behalf of

Dr. Mark Wing Loong Cheong 

Academic Editor

PLOS ONE